# Establishing Task Scaling Laws via Compute-Efficient Model Ladders

**Akshita Bhagia**[*][†]     **Jiacheng Liu**[*♠][†]     **Alexander Wettig**[♣][†]     **David Heineman**[†]

**Oyvind Tafjord**[†]     **Ananya Harsh Jha**[♠][†]     **Luca Soldaini**[†]     **Noah A. Smith**[†♠]

**Dirk Groeneveld**[†]     **Pang Wei Koh**[†♠]     **Jesse Dodge**[†]     **Hannaneh Hajishirzi**[†♠]

[†]Allen Institute for AI     [♠]University of Washington     [♣]Princeton University
[*]Equal contribution.
{akshitab,jiachengl}@allenai.org

## Abstract

We develop task scaling laws and model ladders to predict the **individual task performance** of pretrained language models (LMs) in the **overtrained setting**. Standard power laws for language modeling loss cannot accurately model task performance. Therefore, we leverage a two-step prediction approach: (1) use model and data size to predict an *intermediate loss*, then (2) use it to predict task performance. We train a set of small-scale "ladder" models, collect data points to fit the parameterized functions of the two prediction steps, and make predictions for two target models: a 7B model trained to 4T tokens and a 13B model trained to 5T tokens. Training the ladder models only costs **1% of the compute** used for the target models. On four multiple-choice tasks formatted as ranked classification, we can predict the accuracy of both target models within 2 points of absolute error. We find that tasks with higher prediction error also have higher variance in the metrics over model checkpoints. We also contrast multiple **design choices** for predicting accuracy, and present recommendations for extending our method to new models and tasks.

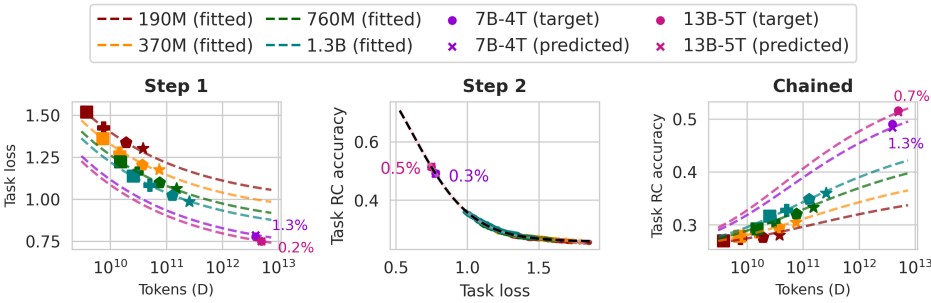

Figure 1: Predicting MMLU accuracy with our method. We use model size $N$ and data size $D$ to predict a "task loss" on MMLU (step 1), and then use this task loss to predict task accuracy in ranked classification format (step 2). The chained plot shows end-to-end prediction from $(N, D)$ to task accuracy. The functions in step 1 and 2 are fitted on data points collected from *ladder* models (markers colored in red, orange, green and cyan); 7B-4T and 13B-5T are the target models for which we make predictions. We report relative prediction error in the plot next to each target model point.

# 1 Introduction

Language models (LMs) are expensive to train. Building a good LM requires a wide range of experimentation over data and architecture choices. Scaling laws, which use the performance of smaller models to predict the performance of a larger model without actually training it, enable more efficient resource allocation for such experimentation. Further, modern LMs are compared based on their performance on *downstream* tasks, rather than perplexity. Certain downstream tasks (e.g., HellaSwag) are also often used as development benchmarks for creating training recipes and good data mixtures for training large LMs (OLMo et al., 2024; Li et al., 2024). Scaling laws for downstream tasks are, therefore, important for LM pretraining.

Prior work has shown results on predicting the average top-1 error over many tasks (as opposed to predicting task accuracy directly) (Gadre et al., 2024), or the accuracy for only one specific task (ARC-Challenge) (Dubey et al., 2024), using at least 5% of the compute required to train the target models. Additionally, these require developing a testbed of many small models to first find compute-optimal models, which adds significant compute costs. Predicting LM performance on a range of individual downstream tasks in a computationally efficient way (e.g., without finding compute-optimal models) remains an open problem.

Here, we tackle the challenge of predicting **individual task performance** of LMs as a function of model size and training data size. In particular, we predict the performance of 7B and 13B models. We focus on a **range of multiple-choice tasks** and predict the task accuracy for problems written in the ranked classification (RC) format. We use a set of small models (190M to 1.3B non-embedding parameters), which we train for varying durations (1x to 10x Chinchilla optimal data size) to develop scaling laws. Training these **fixed-size ladder models** costs only 1% of the compute of the two target models combined.

We employ a two-step approach to (1) use the number of model parameters $N$ and training tokens $D$ (the *input features*) to predict a task-specific loss (an *intermediate feature*), and then (2) use this task loss to predict *accuracy*. We experiment with different **design choices** for the intermediate feature, as well as input features, and apply our method on 8 selected tasks from OLMES (Gu et al., 2024). We quantify the predictability of the tasks based on the variance of target metrics for a representative small model over the last few training checkpoints. Finally, we provide recommendations for developing downstream scaling laws for new models, and selecting a design choice for a given task. From our recommendations, using a task-specific loss as the intermediate feature, we achieve an absolute error of $< 2$ points for our target models on four tasks (MMLU, HellaSwag, PiQA, SocialIQA), and an average absolute error of 4 points across both target models and all tasks.

# 2 Setup

We aim to predict the task performance of LMs with an arbitrary *training scale* – the combination of model size ($N$) and number of training tokens ($D$). As recent LMs are overtrained (Dubey et al., 2024; Li et al., 2024; Bai et al., 2023), we do not constrain $N$ and $D$ to stay close to the compute-optimal regime (Hoffmann et al., 2022). We validate our predictions on the OLMo 2 models after stage 1 pretraining and before stage 2 annealing (OLMo et al., 2024); a 7B model trained to 4T tokens ("**7B-4T**") and a 13B model trained to 5T tokens ("**13B-5T**"), both trained from scratch with the same data mixture. These are *target models*.

## 2.1 Ladder models

To predict the performance of large models, we extrapolate from data points collected from training many small models ("**ladder models**"). These models have the same architecture and are trained with the same data mix as the target models. This is useful for guiding pretraining development, as data mixtures and modeling decisions can be tested in a controlled setting. The models span a relatively wide range of model size and training data size, while collectively costing a small fraction of compute used to train the target models.

Table 1:  Model setup. Up to 1.3B are ladder models; 7B-4T and 13B-5T are target models.

| | 190M | 370M | 760M | 1.3B | 7B-4T | 13B-5T |
|---|---|---|---|---|---|---|
| Model size ($N$) | 190,354,176 | 371,262,464 | 758,220,288 | 1,279,395,840 | 6,887,575,552 | 13,202,396,160 |
| 1xC data size ($D$) | 3,807,083,520 | 7,425,249,280 | 15,164,405,760 | 25,587,916,800 | – | – |
| Batch size (sequences) | 128 | 192 | 320 | 384 | 1,024 | 2,048 |
| Batch size (tokens) | 524,288 | 786,432 | 1,310,720 | 1,572,864 | 4,194,304 | 8,388,608 |
| Training steps (1xC) | 7,272 | 9,452 | 11,580 | 16,279 | – | – |
| Peak LR | $9.7 \times 10^{-4}$ | $7.8 \times 10^{-4}$ | $6.1 \times 10^{-4}$ | $5.2 \times 10^{-4}$ | $3.0 \times 10^{-4}$ | $9.0 \times 10^{-4}$ |
| Warmup steps | 363 | 472 | 578 | 813 | 2000 | 1000 |
| Dimension | 768 | 1,024 | 1,536 | 2,048 | 4,096 | 5,120 |
| Num heads | 12 | 16 | 16 | 16 | 32 | 40 |
| Num layers | 12 | 16 | 16 | 16 | 32 | 40 |
| MLP ratio | 8 | 8 | 8 | 8 | 5.375 | 4 |

Table 2:  Example for RC format from HellaSwag (Zellers et al., 2019). Tokens on which task loss is computed are marked in green.

| Original problem | A woman is outside with a bucket and a dog. The dog is running around trying to avoid a bath. She
A. rinses the bucket off with soap and blow dry the dog's head.
B. uses a hose to keep it from getting soapy.
C. gets the dog wet, then it runs away again.
D. gets into a bath tub with the dog.
Answer: C |
|---|---|
| Task loss calculation | Question: A woman is outside with a bucket and a dog. The dog is running around trying to avoid a bath. She
Answer: gets the dog wet, then it runs away again. |

We define four model sizes $N \in \{$ 190M, 370M, 760M, 1.3B $\}$ (considering only non-embedding parameters) by varying the width and depth of the transformer. We use $D = 20 \cdot N$ as the Chinchilla optimal setting (Hoffmann et al., 2022) (denoted "1xC"), and train each model with number of tokens $D \in \{$ 1xC, 2xC, 5xC, 10xC $\}$. In total, we train $4 \times 4 = 16$ models. We save intermediate checkpoints every 200 steps for 1xC and 2xC runs, every 500 steps for 5xC runs, and every 1000 steps for 10xC runs. Table 1 lists the configurations of each model size.

The target models are trained with a cosine LR schedule with linear warmup, where the LR decays to 10% of the peak LR over a horizon of 5T tokens. We match this in the ladder models, where the decay horizon is adjusted to match the training data size of each model. Unfortunately, the OLMo 2 7B model was only trained to 3.9T tokens in stage 1 pretraining, whereas the cosine scheduler was set to 5T tokens. To account for this incomplete training, we train this model with an additional 50B tokens where we linearly decay the LR down to 10%. Our target 7B-4T model is thus trained on a total of 3.95T tokens, where the linear decay instead of the slower cosine decay allows us to use fewer tokens.

For the ladder models, we set the peak LR, batch size, and warmup steps by extrapolating from the configuration of 7B-4T using the method introduced in Porian et al. (2024). All other configurations follow from the 7B-4T model.

**Compute cost.** Using $C \approx 6ND$ to estimate the compute (Kaplan et al., 2020), the total amount of compute used for training the ladder models is $5.2 \times 10^{21}$ FLOPs. This is only **3.2%** of that used for training the 7B-4T model ($1.6 \times 10^{23}$ FLOPs), **1.3%** of the 13B-5T model ($3.9 \times 10^{23}$ FLOPs), and less than **1.0%** of both target models combined.

## 2.2   Intermediate features and accuracy

**Task loss.** We define task loss as the negative log-likelihood of the correct answer sequence, divided by its length in bytes, also known as the bits-per-byte (bpb) metric. Table 2 shows the problem formatting for computing task loss on an example. We use bpb instead of normalizing by number of tokens to reduce the impact of the tokenizer.[1]

**TaskCE.** Task cross-entropy loss which accounts for incorrect answers (§C.2).

---

[1]similar to "normalized NLL per char" (Dubey et al., 2024).

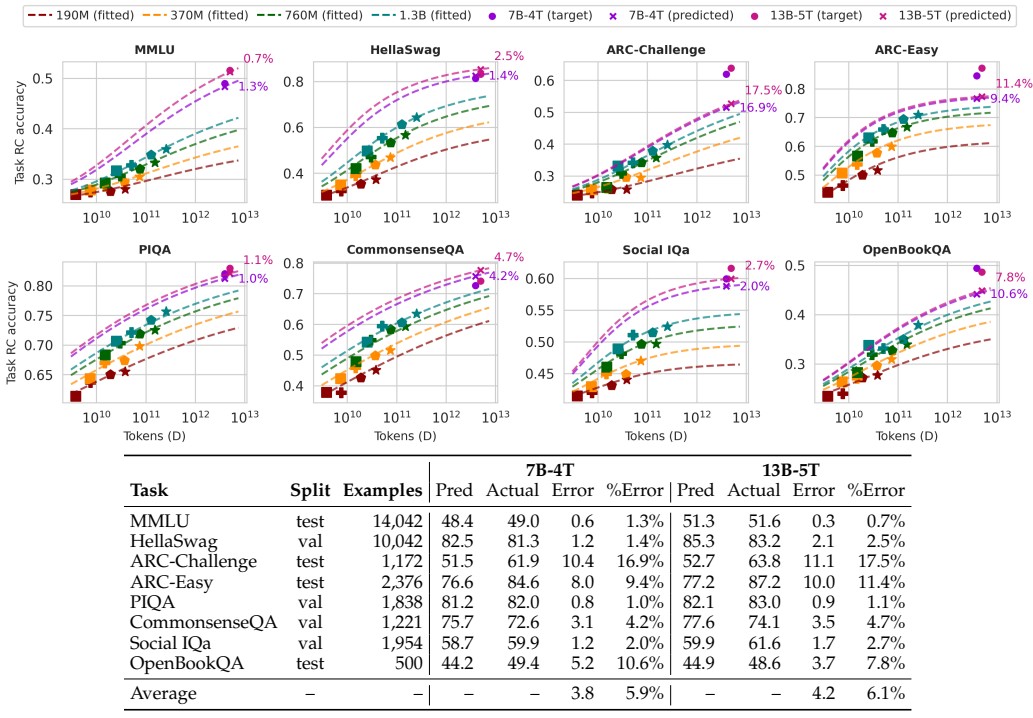

| Task | Split | Examples | 7B-4T | | | | 13B-5T | | | |
|------|-------|----------|-------|--------|-------|-------|-------|--------|-------|-------|
| | | | Pred | Actual | Error | %Error | Pred | Actual | Error | %Error |
| MMLU | test | 14,042 | 48.4 | 49.0 | 0.6 | 1.3% | 51.3 | 51.6 | 0.3 | 0.7% |
| HellaSwag | val | 10,042 | 82.5 | 81.3 | 1.2 | 1.4% | 85.3 | 83.2 | 2.1 | 2.5% |
| ARC-Challenge | test | 1,172 | 51.5 | 61.9 | 10.4 | 16.9% | 52.7 | 63.8 | 11.1 | 17.5% |
| ARC-Easy | test | 2,376 | 76.6 | 84.6 | 8.0 | 9.4% | 77.2 | 87.2 | 10.0 | 11.4% |
| PIQA | val | 1,838 | 81.2 | 82.0 | 0.8 | 1.0% | 82.1 | 83.0 | 0.9 | 1.1% |
| CommonsenseQA | val | 1,221 | 75.7 | 72.6 | 3.1 | 4.2% | 77.6 | 74.1 | 3.5 | 4.7% |
| Social IQa | val | 1,954 | 58.7 | 59.9 | 1.2 | 2.0% | 59.9 | 61.6 | 1.7 | 2.7% |
| OpenBookQA | test | 500 | 44.2 | 49.4 | 5.2 | 10.6% | 44.9 | 48.6 | 3.7 | 7.8% |
| Average | – | – | – | – | 3.8 | 5.9% | – | – | 4.2 | 6.1% |

Figure 2: Task accuracy prediction for the target models using task loss as the intermediate feature. ■ = 1xC; ✚ = 2xC; ⬟ = 5xC; ★ = 10xC. Prediction error is next to the target.

**LM loss.** Standard language modeling loss on the C4-en validation set (Raffel et al., 2019).

**Task accuracy.** Ranked classification (RC) and multiple-choice (MC) are two different *formats* to pose multiple-choice problems.) In RC, the predicted answer is the one with the minimum task loss. In MC, all the answer choices are included in the prompt, and the predicted answer is the answer code (e.g., A, B, C, etc.) with the smallest loss (Gu et al., 2024). Here, we focus on the RC format.[2]

**Evaluation.** Following Gadre et al. (2024), we compute the relative error for the task accuracy to evaluate the goodness of prediction, defined as

$$\text{Relative Error} = \frac{|\text{prediction} - \text{actual}|}{\text{actual}} \times 100\%.$$

### 2.3 Task selection

We include the following 8 tasks from the OLMES evaluation suite (Gu et al., 2024): MMLU (Hendrycks et al., 2021), HellaSwag (Zellers et al., 2019), ARC-Challenge (Clark et al., 2018), ARC-Easy (Clark et al., 2018), PIQA (Bisk et al., 2020), CommonsenseQA (Talmor et al., 2019), Social IQa (Sap et al., 2019), and OpenBookQA (Mihaylov et al., 2018). We exclude BoolQ (Clark et al., 2019) and Winogrande (Sakaguchi et al., 2020), as the task loss and accuracy are noisier for these tasks. §5 discusses task predictability further. We use the test set where possible, and fall back to the validation set otherwise. For all tasks, we use the 5-shot setting provided in OLMES. Figure 2 shows the task statistics.

---

[2]RC reliably measures progress for models across a wide range of scales, whereas MC capability does not emerge until the model has several billion parameters.

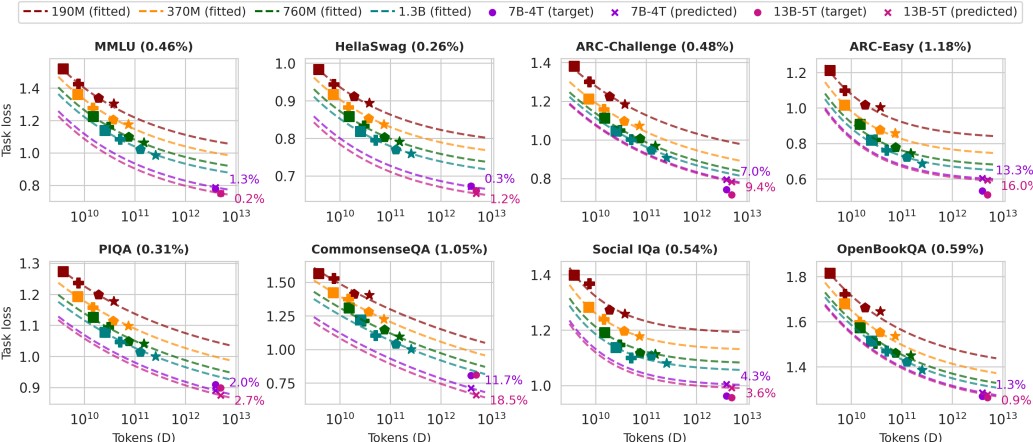

Figure 3: Task loss vs training scale $(N, D)$, with fitting on the power function in Equation 1. ■ = 1xC; ✚ = 2xC; ⬟ = 5xC; ★ = 10xC. We report the average relative fitting error in parentheses after the task name, and prediction error next to the target model point.

## 3  Method

We break down task accuracy prediction into two steps: 1) predicting the intermediate feature (we use task loss to illustrate this), and 2) using it to predict the task accuracy.

### 3.1  Step 1: Use $(N, D)$ to predict the intermediate feature

We consider three intermediate features: task loss, taskCE loss, and LM loss (defined in §2).

As proposed by Hoffmann et al. (2022) and followed by others (Muennighoff et al., 2023; Gadre et al., 2024; Zhang et al., 2024), the LM loss of a model on a held-out eval set can be modeled as a power function with respect to $N$ and $D$:

$$L(N, D) = A/N^{\alpha} + B/D^{\beta} + E, \tag{1}$$

where $A, B, \alpha, \beta, E$ are parameters to fit.

We postulate that **the same functional form applies to these losses**. We validate this assumption by looking at the goodness of function fitting.

**Function fitting.** We take the loss value of the final checkpoint of each ladder model $\{(N_i, D_i, L_i)\}_{i=1}^{n}$ (where $n = 16$), and use this data to fit the parameters of Equation 1. We fit a separate set of parameters for each task. Following Hoffmann et al. (2022), we minimize the Huber loss between the logarithm of predicted and actual loss: $\frac{1}{n} \sum_{i=1}^{n} \text{Huber}_{\delta} \left( \log \hat{L}(N_i, D_i) - \log L_i \right)$, where $\delta = 10^{-3}$. We optimize $A$ and $B$ in log space – we apply transformation $a = \log A, b = \log B$ and optimize $(a, b, \alpha, \beta, E)$. We use the L-BFGS-B optimizer as implemented in `scipy.minimize()`, with constraints $A, B, \alpha, \beta, E \geq 0$; with these constraints, the function is convex, and thus L-BFGS-B is guaranteed to converge.[3]

We take the average loss over the last 5 checkpoints of each ladder model to reduce noise.

**Results preview.** Figure 3 shows the function fitting of step 1 on the ladder models, and the prediction of task loss as the intermediate feature for the target models. The power function gives an average relative *fitting error* ranging from 0.2% to 1.2%, which suggests that Equation 1 is a good functional form to describe task losses. We have a relative *prediction error* within 3% on MMLU, HellaSwag, PIQA, and OpenBookQA. The method underestimates the loss for CSQA, and overestimates on ARC-Challenge, ARC-Easy, and Social IQa. On

---

[3]To test if this function is convex, we confirm that the Hessian matrix of second derivatives is positive semi-definite everywhere.

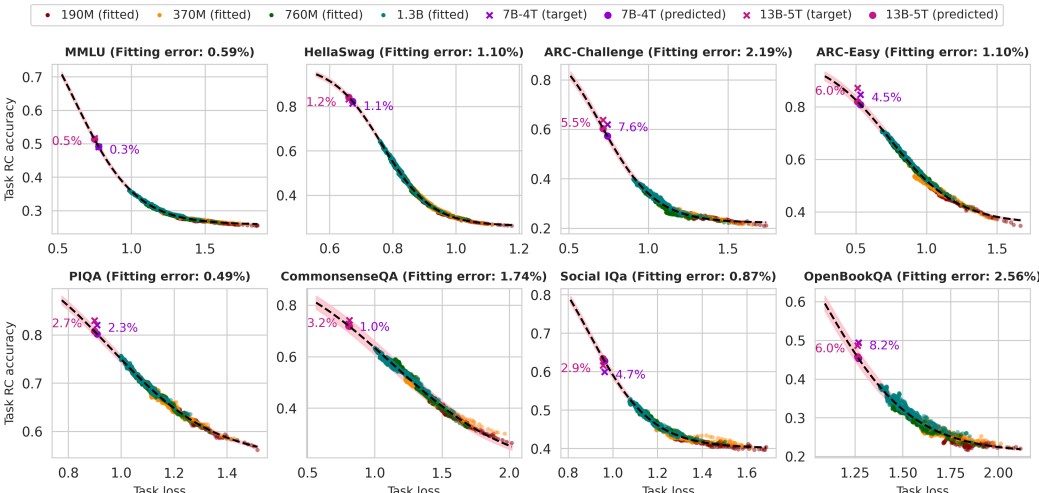

Figure 4: Task RC accuracy vs task loss, with fitting on the sigmoid function in Equation 2. The prediction error is next to the target model point. The shaded area represents the prediction intervals for the fitted function.

target models, across all tasks we get an average relative error of 5.2% for 7B-4T, and 6.6% for 13B-5T. Results for the other intermediate features are discussed in §C.2 and §C.3.

In §C.1, we use compute-FLOPs as input variable instead of $(N, D)$, and observe higher fitting errors for all tasks. We also note that FLOPs cannot distinguish between compute-optimal and overtrained models.

## 3.2 Step 2: Use intermediate feature to predict accuracy

Following Dubey et al. (2024), we model the mapping from the intermediate loss to accuracy with a sigmoidal function:

$$Acc(L) = \frac{a}{1 + e^{-k(L-L_0)}} + b \tag{2}$$

where $a, b, k, L_0$ are parameters to fit.

The choice of a sigmoidal functional form is motivated as follows: a weak model has high task loss and random task accuracy, and a strong model has low task loss and high task accuracy that saturates at 100%. We observe (as shown in Figure 4) that the $(L_i, Acc_i)$ points collected from different ladder models tend to fall on a shared sigmoidal curve; this applies to both intermediate and final model checkpoints.

**Function fitting.** We use the intermediate loss and accuracy values from both final and intermediate checkpoints of the ladder models $\{(L_i, Acc_i)\}_{i=1}^m$ (where $m \approx 1400$) to fit the parameters of Equation 2. We fit a separate set of parameters for each downstream task. We minimize the L2 loss between the predicted and actual accuracy: $\frac{1}{m} \sum_{i=1}^m (\hat{Acc}(L_i) - Acc_i)^2$. We use non-linear least squares implemented by `scipy.optimize.curve_fit()` to fit this equation, as sigmoid functions are not convex.

The variation from checkpoint to checkpoint can be high. To smoothen the noise, we apply a moving average on the intermediate loss and task accuracy over all checkpoints of each training run, with a window size of 5. We also discard the checkpoints from the first 10% of each training run as these are quite noisy, and add an extra data point ($L = 0.0, Acc = 1.0$) to the training pairs. This helps avoid the cases when the fitting function degenerates for very noisy data, and the ladder models are too small to do well on certain tasks.

**Results preview.** Figure 4 shows the function fitting of step 2 on the ladder models, and the prediction of task accuracy for the target models using their *actual* task loss as the intermediate feature. For all tasks, all data points fit well with a shared sigmoidal function,

Table 3: Comparison of design choices. **Upper:** Average step 1 prediction error. **Lower:** Average chained prediction error. For MMLU, using task loss yields the lowest prediction error. We observe higher errors for ARC-C, ARC-E, and OBQA with task loss, which aligns with our variance analysis in §5, and that C4 as the intermediate loss works better for them.

| Design choice | 7B-4T | | | | | | | | 13B-5T | | | | | | | |
|---|---|---|---|---|---|---|---|---|---|---|---|---|---|---|---|---|
| | MMLU | HS | ARC-C | ARC-E | PIQA | CSQA | SIQa | OBQA | MMLU | HS | ARC-C | ARC-E | PIQA | CSQA | SIQa | OBQA |
| (N, D) → task loss (§3) | 1.3% | 0.3% | 7.0% | 13.3% | 2.0% | 11.7% | 4.3% | 1.3% | 0.2% | 1.2% | 9.4% | 16.0% | 2.7% | 18.5% | 3.6% | 0.9% |
| FLOPs → task loss (§C.1) | 4.3% | 2.1% | 2.4% | 5.3% | 2.0% | 12.5% | 5.7% | 0.5% | 5.2% | 2.6% | 3.5% | 7.0% | 2.6% | 18.8% | 5.7% | 1.4% |

| Design choice | 7B-4T | | | | | | | | 13B-5T | | | | | | | |
|---|---|---|---|---|---|---|---|---|---|---|---|---|---|---|---|---|
| | MMLU | HS | ARC-C | ARC-E | PIQA | CSQA | SIQa | OBQA | MMLU | HS | ARC-C | ARC-E | PIQA | CSQA | SIQa | OBQA |
| (N, D) → task loss (§3) | **1.3%** | 1.4% | 16.9% | 9.4% | 1.0% | 4.2% | 2.0% | 10.6% | **0.7%** | 2.5% | 17.5% | 11.4% | 1.1% | 4.7% | 2.7% | 7.8% |
| TaskCE (§C.2) | 18.3% | 7.2% | 21.3% | 5.4% | 3.1% | **2.6%** | **0.7%** | 7.1% | 20.2% | 10.5% | 19.5% | 6.2% | 2.9% | **2.7%** | **1.2%** | **0.2%** |
| C4 loss (§C.3) | 2.2% | 4.5% | **1.4%** | **1.2%** | **0.7%** | 5.8% | 6.2% | **2.0%** | 5.0% | 5.7% | **3.9%** | **1.9%** | 1.5% | 7.0% | 7.6% | 10.4% |
| Single step (§C.5) | 5.6% | **0.5%** | 21.7% | 6.1% | **0.7%** | 5.1% | 5.4% | 9.0% | 7.1% | **0.8%** | 22.6% | 7.9% | **0.6%** | 6.6% | 6.8% | 4.7% |

with the average relative *fitting error* ranging from 0.4% to 2.6%. We get within 3% relative *prediction error* on MMLU, HellaSwag, PIQA, and CommonsenseQA. Overall, we get an average relative error of 3.7% for 7B-4T, and 3.5% for 13B-5T.

### 3.3 Chaining the two steps

We chain the two steps by first predicting the intermediate loss with the fitted function in step 1, and then inserting it into the fitted function in step 2 to predict the task accuracy.

**Results preview.** Using task loss as the intermediate feature, we get an average absolute error of 3.8 points for 7B-4T, and 4.2 points for 13B-5T. In §C.5, we also show the results of predicting the task accuracy directly from (N, D) in a single step, and observe higher average prediction errors for both target models (> 30 points).

## 4 Results

Table 3 compares prediction errors for all design choices.

**Task loss.** Figure 2 shows the chained prediction results on the target models using task loss as the intermediate feature. On four tasks – MMLU, HellaSwag, PIQA, and Social IQa – we predict the accuracy within an absolute error of 2 points. For ARC-C and ARC-E, we overestimate the task loss in step 1, and underestimate the task accuracy in step 2. In §5, we analyze the ability of the ladder to predict task performance by considering variation between checkpoints. We find that our results here track with the variation analysis (ARC-E and ARC-C display higher variance).

**TaskCE.** Using TaskCE as the intermediate feature results in overall higher prediction errors for several tasks (including MMLU and HellaSwag; see §6 on why these tasks are especially important). Full results using TaskCE are shown in §C.2.

**LM loss.** Interestingly, the loss on C4-en validation set is a good predictor of task accuracy for several tasks. One possible explanation for this is potential domain overlap between the task and the validation set. We discuss pros and cons of using LM loss in §6. Full results using LM Loss are show in §C.3.

Figure 16 compares absolute and relative errors for all three intermediate features.

## 5 Analysis: Task predictability with the ladder

Some tasks are inherently more challenging to predict reliably with the ladder models; for example, a test set with an inadequate sample size, low-quality test instances or questions that are too difficult for small models. We anticipate three sources of prediction failure:

- High variance in intermediate loss; less reliable data points for function fitting.

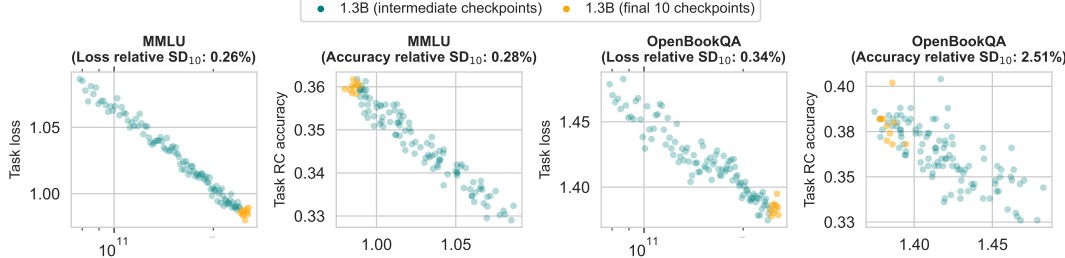

Figure 5: Relative SD over the final 10 checkpoints ($SD_{10}$) for the 1B-10xC ladder model on MMLU and OpenBookQA. OBQA metrics appear noisier than MMLU, resulting in a higher $SD_{10}$. Tasks with high intermediate checkpoint noise indicate higher prediction error (Table 4). Results across all tasks are in Figure 6.

Table 4: Absolute and relative standard deviation of last 10 training checkpoints ($SD_{10}$) for the 1B-10xC model on task loss and accuracy, and relative error for predicting the 7B-4T model on task loss (step 1 only), accuracy (step 2 only) and chained accuracy (step 1 and step 2). We observe that tasks with above-average $SD_{10}$ for 1B-10xC, highlighted in red, tend to have high prediction errors for the 7B-4T model. In particular, we observe a strong Pearson correlation between loss $SD_{10}$ and accuracy prediction error ($r = 0.821$, $p = 0.004$).

| Task | Std. dev. of final 1B checkpoints | | | | Predictions for 7B-4T | | |
| | Loss $SD_{10}$ | Loss % $SD_{10}$ | Accuracy $SD_{10}$ | Accuracy % $SD_{10}$ | Loss % Error | Accuracy % Error | Chained % Error |
|---|---|---|---|---|---|---|---|
| Winogrande | 0.0115 | 0.75 % | 0.0048 | 0.77 % | 4.5 % | 23.7 % | 8.9 % |
| BoolQ | 0.0068 | 1.76 % | 0.0186 | 2.86 % | 11.4 % | 4.3 % | 1.8 % |
| CommonsenseQA | 0.0056 | 0.56 % | 0.0035 | 0.55 % | 11.7 % | 1.0 % | 4.2 % |
| OpenBookQA | 0.0047 | 0.34 % | 0.0095 | 2.51 % | 1.3 % | 8.2 % | 10.6 % |
| ARC-Easy | 0.0045 | 0.66 % | 0.0043 | 0.61 % | 13.3 % | 4.5 % | 9.4 % |
| ARC-Challenge | 0.0037 | 0.40 % | 0.0040 | 1.00 % | 7.0 % | 7.6 % | 16.9 % |
| MMLU | 0.0026 | 0.26 % | 0.0010 | 0.28 % | 1.3 % | 0.3 % | 1.3 % |
| Social IQa | 0.0024 | 0.23 % | 0.0032 | 0.61 % | 4.3 % | 4.7 % | 2.0 % |
| PIQA | 0.0019 | 0.19 % | 0.0024 | 0.31 % | 2.0 % | 2.3 % | 1.0 % |
| HellaSwag | 0.0007 | 0.09 % | 0.0016 | 0.25 % | 0.3 % | 1.1 % | 1.4 % |

- High variance in task accuracy; higher spread of prediction targets.
- Random-chance task accuracy in smaller ladder models due to task difficulty.

We try to determine which tasks will exhibit high prediction errors, using task loss as the intermediate feature. We use the intermediate checkpoints of the largest ladder model (1B-1xC) to measure the noise for task loss and task accuracy. Failure due to random-chance accuracy is discussed when predicting multiple-choice tasks in §B.2.

**Variance analysis.** We compute the standard deviation of the last $n$ training checkpoints of 1B-10xC ($SD_n$). We also compute relative SD (also called the coefficient of variation) to compare between tasks:

$$\text{Relative SD}_n = \frac{\text{SD (final } n \text{ checkpoints)}}{\text{Mean (final } n \text{ checkpoints)}} \times 100\% \qquad (3)$$

A task where ladder models exhibit a higher standard deviation across adjacent evaluated checkpoints potentially indicates a higher prediction error for the target models. To illustrate $SD_n$, we show the intermediate checkpoints for the largest ladder model on OpenBookQA and MMLU in Figure 5, where we find that $SD_{10}$ captures the apparent noise between adjacent training checkpoints.

We present $SD_{10}$ for the 1B-10xC model Table 4, alongside prediction errors for the 7B-4T model. Benchmarks with low $SD_{10}$ (e.g. MMLU and HellaSwag), also have low prediction

errors for the target, indicating these tasks are easier for the ladder to predict. We also find that Loss $SD_{10}$ is correlated with step 2 accuracy error (Pearson $r = 0.821, p = 0.004$ for 7B-4T and $r = 0.855, p = 0.002$ for 13B-5T). Thus, $SD_{10}$ can be reported alongside predictions to explain which benchmarks may have high error before running the target model.

## 6    Discussion and Recommendations

In §4, we observed that different design choices work well for different tasks. Here, we discuss factors that affect final predictions, and present some guidelines to develop task scaling laws for a new overtrained model or new task (Figure 16 illustrates these findings):

**Target variance.**    Variance analysis is useful for determining if it is feasible to predict the target metrics (both intermediate loss and accuracy). For tasks with high variance, especially for task accuracy, we generally expect to see lower predictability. High variance in a particular intermediate loss can potentially be mitigated with alternative intermediate features. For new tasks, **variance analysis should be conducted on the largest ladder model** that can be trained with available compute (we used 1B-10xC).

**Choice of intermediate features.**    It is important for the intermediate feature to be both 1) predictable, and 2) a good predictor for task accuracy. Tasks with a noisy mapping between their intermediate feature and accuracy (e.g., ARC-C, ARC-E and OpenBookQA) may have compounded errors in the two-step prediction. For ARC-C and OpenBookQA in particular, their high variance for both the intermediate task loss and accuracy make them challenging to predict with the ladder. HellaSwag and PIQA, the tasks where we observe the least variance, are the easiest to predict across all design choices, including the combined single step approach. We also note that while LM loss on C4 is a reasonable predictor of task accuracy for several tasks in our setup, this may not hold for other novel downstream tasks on different domains. As a general rule, using **a task-specific loss may work across a wider range of downstream tasks**.

**Number of task instances.**    Tasks with fewer instances show inconsistent results. E.g. OpenBookQA has only 500 instances and has a high prediction error for most design choices. Even using LM loss as the intermediate feature, while the prediction error is low, the fitting error is actually high (2.35%) as shown in §C.3, indicating randomness. On the other hand, MMLU and HellaSwag have large sample sizes – 5x larger than the other tasks. Given their low sample variance and noise around the prediction target, we expect that their errors are most representative of the true difficulty of predicting downstream performance, so even though taskCE and task loss perform comparably for several tasks, task loss is better considering task size. If the goal is to select a single design choice that works for all tasks, the **design choice should be validated on tasks with a large number of instances**.

**Robustness of the ladder models.**    In §D, we predict the task accuracies for a larger OLMo 2 target model: 32B-6T, using the same ladder models (0.45% of the target model compute). While the absolute errors are higher, they still follow the same trend as the smaller target models (lower variance tasks exhibit relatively lower prediction errors), indicating that **the same ladder can reliably be used to estimate the performance of much larger models**.

## 7    Related Work

**Scaling laws for language modeling.**    Kaplan et al. (2020) and Hoffmann et al. (2022) were among the first to postulate the functional form of LM losses as a power function of model parameters and data size. The Chinchilla equation, in particular, has become the basis of many subsequent scaling law works (Muennighoff et al., 2023; Gadre et al., 2024; Zhang et al., 2024). This line of work focuses on predicting the LM loss (rather than downstream performance) on a held-out set with similar distribution as the pretraining data, but not on downstream tasks which is more important and challenging.

**Scaling laws for downstream tasks.**    Scaling laws for downstream tasks have been explored in Gadre et al. (2024). Instead of directly predicting accuracy for individual tasks,

they compute the average top-1 error over 17 LLM-foundry (MosaicML, 2024) evaluation tasks as a function of cross entropy loss on C4 (Raffel et al., 2019). Dubey et al. (2024) uses a two-step prediction to first map the the training compute to the negative log-likelihood of the correct answer for a single task in an evaluation benchmark, and then relate the log-likelihood to the task accuracy. Unlike our work, which uses a fixed ladder of small models, they rely on first finding compute-optimal models. Chen et al. (2024) also employs a two-stage approach for predicting downstream performance, but uses the pre-training loss instead of a task-specific loss as the intermediate step. Isik et al. (2024) studies scaling laws in the finetuning setup, for machine translation tasks. Polo et al. (2025) leverages models of different families to predict downstream performance, but this leads to less accurate predictions for guiding pretraining development for a specific family of models.

## 8 Conclusion and Future Work

We develop model ladders and task scaling laws to predict downstream task performance of overtrained LMs (7B–4T and 13B–5T). Specifically, our contributions include predicting **individual task performance** for a **range of tasks** with multiple **design choices**; using a **fixed set of ladder models** (rather than using a large set of models to find compute-optimal models for each size); and using a small compute budget (**1% of target compute**). We also conduct variance analysis to determine task predictability, and provide recommendations for picking good design choices.

In future work, we hope to explore ways to improve task predictability, such as increasing the size of evaluation sets, alternative evaluation formats, reducing the impact of randomness, etc. We also hope to extend our prediction method to tasks in the multiple-choice (MC) format to more accurately reflect the capabilities of larger LMs. MC accuracy is harder to extrapolate from our smaller ladder models; we show some preliminary results in §B.2. Finally, we also hope to validate our method on larger models (70B parameters and beyond).

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

## A  Fitted Parameters

In Table 5, we list the parameters of the fitted functions in Figure 3 and Figure 4.

## B  Additional Analyses

### B.1  Additional variance analysis results

Figure 6 shows the intermediate checkpoints for the largest ladder model across all tasks, along with the relative standard deviation over the final 10 checkpoints (% $SD_{10}$). In Table 4, we show that this intermediate checkpoint noise corresponds to the difficulty of predicting the task performance for the target model.

### B.2  Predicting task accuracy under the MC format

In this paper we have been focusing on the ranked classification (RC) format of tasks. Here, we explore if we can make predictions when problems are written in the multiple-choice (MC) format. This is a challenging problem, as small models often exhibit random performance on standard downstream tasks (e.g., MMLU) until a certain size threshold (Hu et al., 2023). Thus, it is not practical to fit the mapping from task loss to MC accuracy in step 2 with data points from the ladder models. Instead, we use data points from early intermediate checkpoints of the target models.

Observing the MC accuracy curves during training of the 7B-4T and 13B-5T models (Figure 7 upper), we find that they have three phases: (1) very early in training, MC accuracy is random; (2) at some point, MC accuracy increases rapidly; (3) finally, it starts growing

| Task | Step 1 Fitted Function |
|------|------------------------|
| MMLU | $L(N, D) = 38.07/N^{0.23} + 100.09/D^{0.24} + 0.45$ |
| HellaSwag | $L(N, D) = 11.23/N^{0.20} + 60.37/D^{0.26} + 0.50$ |
| ARC-Challenge | $L(N, D) = 702974.93/N^{0.79} + 38.45/D^{0.20} + 0.65$ |
| ARC-Easy | $L(N, D) = 79412.07/N^{0.66} + 3957.51/D^{0.42} + 0.56$ |
| PIQA | $L(N, D) = 405.66/N^{0.40} + 10.16/D^{0.15} + 0.72$ |
| CommonsenseQA | $L(N, D) = 56.86/N^{0.23} + 10.91/D^{0.11} + 0.00$ |
| Social IQa | $L(N, D) = 1200.94/N^{0.45} + 7897.19/D^{0.48} + 0.95$ |
| OpenBookQA | $L(N, D) = 86346.32/N^{0.69} + 137.35/D^{0.26} + 1.20$ |

| Task | Step 2 Fitted Function |
|------|------------------------|
| MMLU | $Acc(L) = -0.74/(1 + \exp(-4.83(L - 0.62))) + 1.00$ |
| HellaSwag | $Acc(L) = -0.73/(1 + \exp(-12.74(L - 0.77))) + 0.99$ |
| ARC-Challenge | $Acc(L) = -0.78/(1 + \exp(-5.91(L - 0.71))) + 1.00$ |
| ARC-Easy | $Acc(L) = -0.65/(1 + \exp(-4.13(L - 0.74))) + 1.00$ |
| PIQA | $Acc(L) = -0.46/(1 + \exp(-5.03(L - 0.96))) + 1.00$ |
| CommonsenseQA | $Acc(L) = -0.86/(1 + \exp(-2.21(L - 1.13))) + 1.00$ |
| Social IQa | $Acc(L) = -0.60/(1 + \exp(-7.16(L - 0.89))) + 1.00$ |
| OpenBookQA | $Acc(L) = -0.79/(1 + \exp(-4.31(L - 1.08))) + 1.00$ |

Table 5: Parameters for the fitted functions for Figure 3 and Figure 4.

steadily with more training steps. The phase of rapid growth for a model is strikingly identical across all tasks: around 70k steps for the 7B-4T model, and around 20k steps for the 13B-5T model. As a result, the data points $\{(L_i, Acc_i)\}$ fall on a peculiar shape that cannot be described by a sigmoidal function (Figure 7 lower left).

Therefore, we fit the sigmoidal curve on data points collected from intermediate checkpoints during the third phase of the MC accuracy curve. For 7B-4T, we use steps between 170k and 450k; for 13B-5T, we use steps between 50k and 150k. Noticing that the curves for the two models do not coincide, we fit a function for each model separately (as opposed to a single joint fitting in RC). We conduct a case study with MMLU and show the fitting and prediction in Figure 7 (lower right). Using data points from the third phase, we can reliably extrapolate the mapping from task loss to MC accuracy. We predicted the MC accuracy of 7B-4T with 3.0% relative error, and that of 13B-5T with 1.0% relative error. When chaining this with the step 1 fitted function, our end-to-end prediction of MMLU accuracy has an absolute error of 0.3 points on the 7B-4T model, and 0.4 points on the 13B-5T model.

**Compute overhead.** To fit the step 2 function for MC accuracy, we do need to collect data points from an early part of the target model training. 7B-4T is trained for 928k steps, and we use data points up to 450k, which is about 50% of the full training compute. 13B-5T is trained for 596k steps, and we use data points up to 150k, which is about 25% of the full training compute. We note this is significantly more compute than used in the model ladders for predicting RC performance. However, as noted above, smaller models (which are less compute intensive) don't provide useful signal here. This is an opportunity for future work.

### B.3 Compute requirements for predicting each task

We consider the impact of the scale of the model ladder on the prediction for the target model. In general, the prediction is harder if the difference of scale between the ladder and the target models is larger. We explore this trade-off using the **7B-4T** model as the target.

In Figure 8, we progressively increase the compute-FLOPs used for function fitting. The prediction error is significantly worse with fewer FLOPs. On MMLU, our full ladder (3.2% compute of the target model) gets 1.3% error. Reducing the number of FLOPs to 0.1% of the target model compute increases the error to ~12% which is an order of magnitude higher.

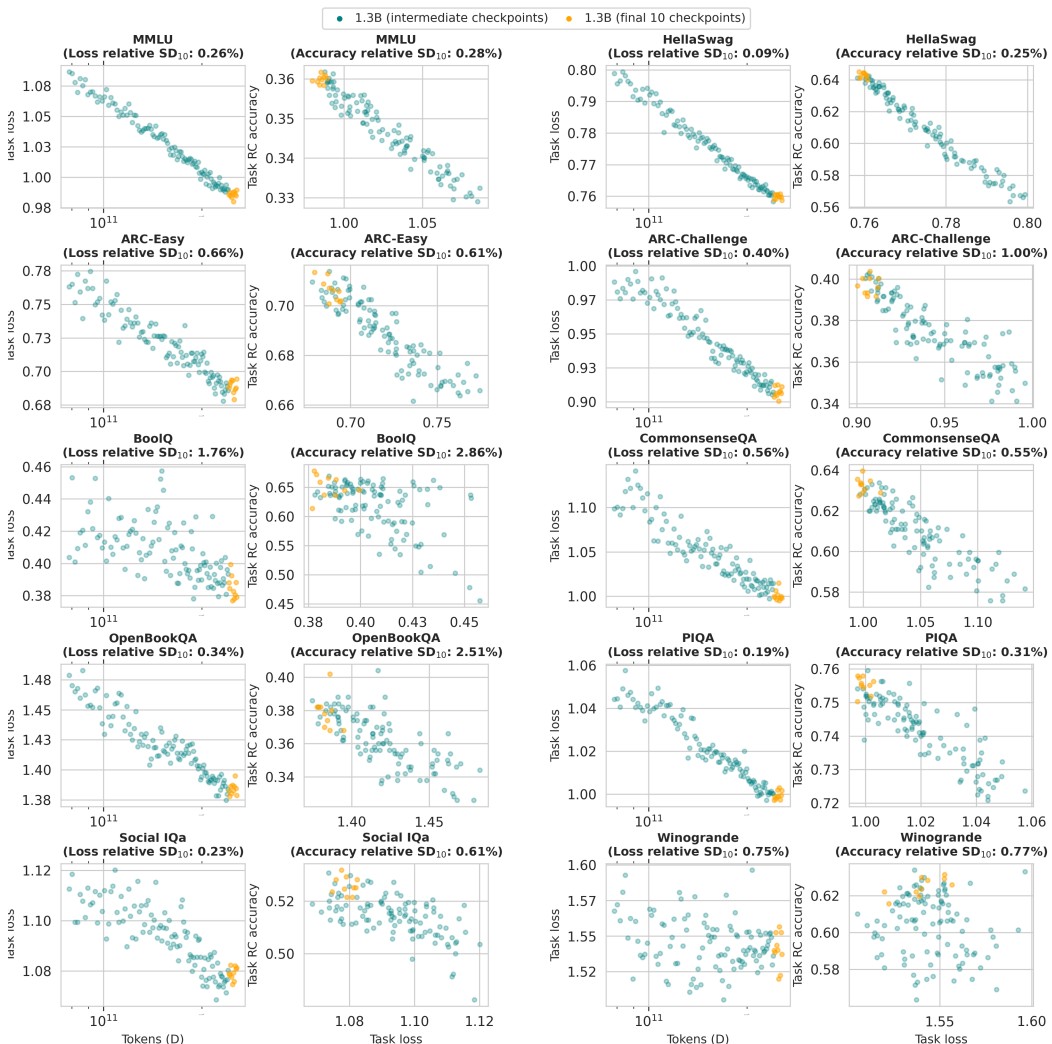

Figure 6: Intermediate checkpoints and standard deviation over the final 10 checkpoints ($SD_{10}$) for the 1B-10xC ladder model. We find some tasks exhibit high noise between adjacent training checkpoints, which is indicative of the inherent difficulty in predicting performance for such tasks using the model ladder.

This is more observable in certain tasks. Interestingly, we see a slight improvement in errors with less compute in ARC-C and ARC-E, potentially due to the variance as seen in §5.

We also consider the impact of each axis of the ladder models (model size $N$ and Chinchilla multiplier $xC$) on the prediction error for each task.

Figure 9 shows the prediction errors for each step as we progressively increase the model size included in the ladder (i.e., ladder upto 760M will include 190M, 370M, 760M). We observe a downward trend in the prediction error for most tasks as the ladder size gets closer to the target model.

Figure 10 similarly shows the impact of including models with longer training regimes. Interestingly, we do not see a significant reduction in the prediction error as we include the ladder models trained to higher Chinchilla multipliers[4]. There is a slight downward trend (noticeable in MMLU), but this dimension has more flexibility.

---

[4]The target model 7B-4T is trained to approximately 28xC.

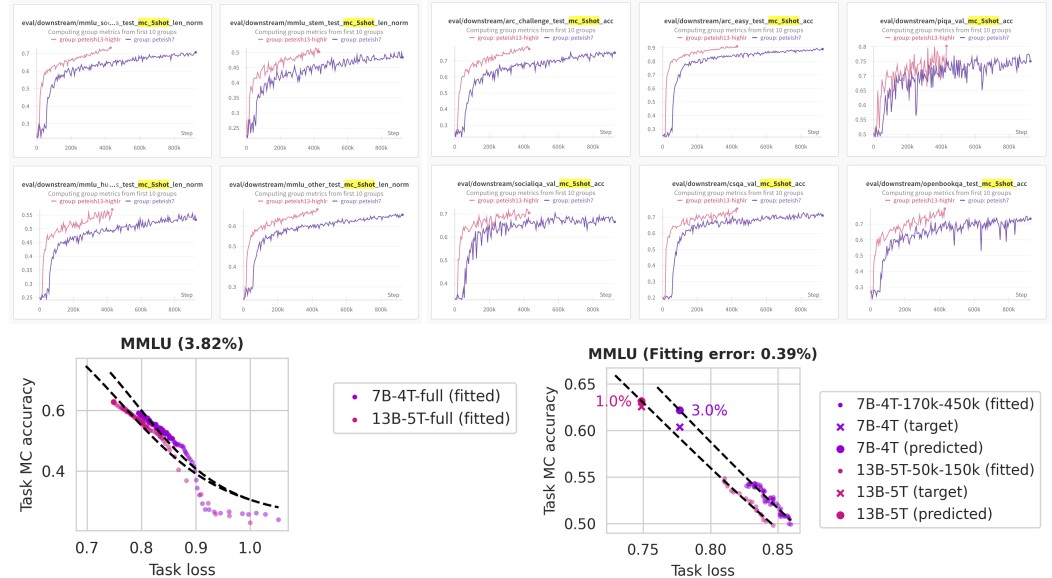

Figure 7: **Upper:** MC accuracy curves during training of 7B-4T and 13B-5T. **Lower left:** Task MC accuracy vs task loss, with data points from all intermediate checkpoints of 7B-4T and 13B-5T. A sigmoidal function cannot fit the data points. **Lower right:** Task MC accuracy vs task loss, where the sigmoidal function (Equation 2) is fitted on data points from the intermediate checkpoints between 170k–450k steps of 7B-4T, and between 50k–150k steps of 13B-5T.

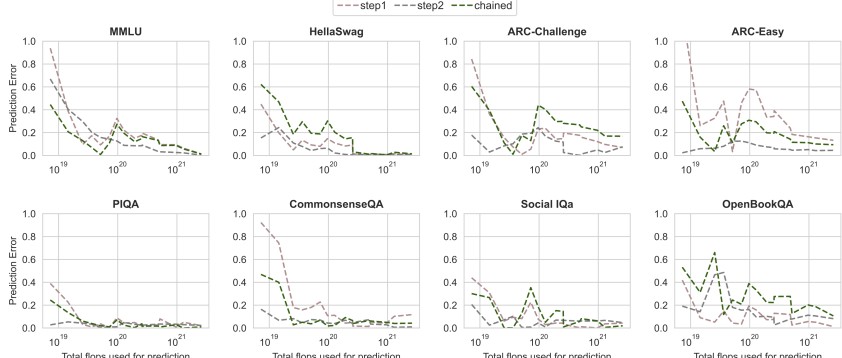

Figure 8: Prediction error on the 7B-4T target model as a function of the total compute-FLOPs used in the ladder models for prediction. The left-most point uses only the smallest model (190M-1xC) for prediction. The right-most point uses the full ladder (all 16 models). The prediction error generally reduces as the ladder FLOPs increase. ARC-C, ARC-E, and OBQA display higher variation (which can be attributed to the variation analysis in §5), but still have a downward trend.

For users of our approach, and future work, we encourage exploration in reducing the overall cost by considering both these dimensions. Concretely, given a fixed budget for the ladder models, increasing the model size $N$ is more likely to improve the prediction error, compared to training for longer.

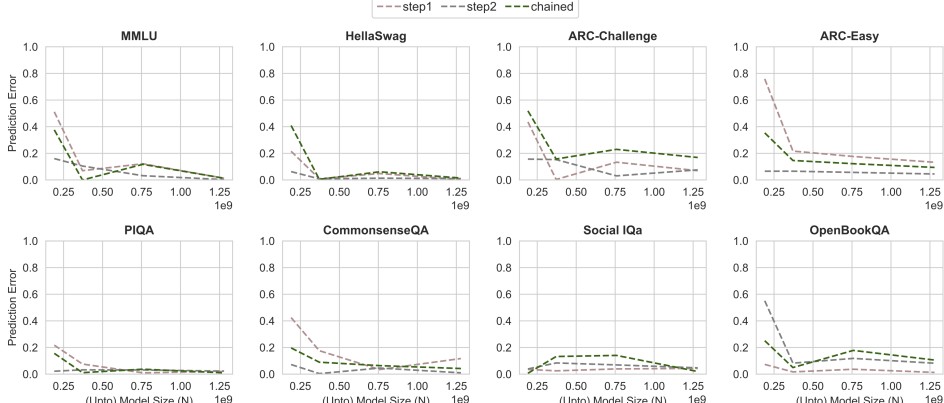

Figure 9: Prediction error on the 7B-4T target model when including **up to** model size ($N$) in the ladder for prediction. Eg. $N$ up to 760M will include 190M, 370M, 760M models trained to 1xC, 2xC, 5xC, 10xC. For most tasks, we observe a downward trend in the prediction error as $N$ increases.

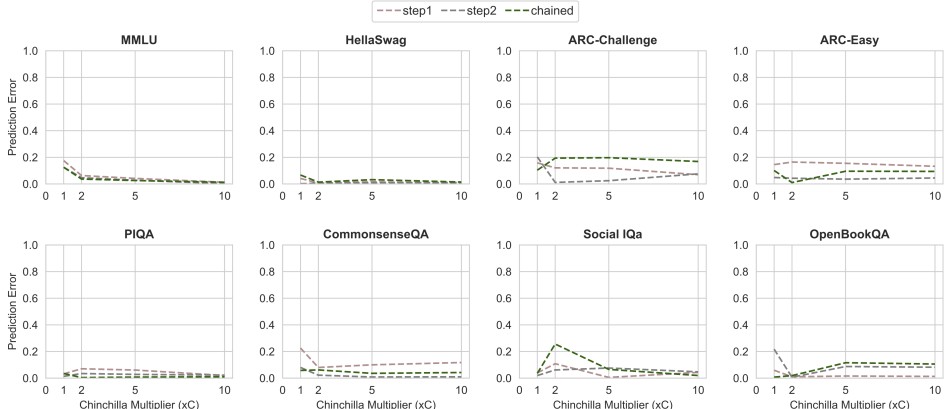

Figure 10: Prediction error on the 7B-4T target model when including models trained **up to** chinchilla multiplier ($xC$). Eg. $xC$ up to 2xC will include 190M, 370M, 760M, 1B models trained to 1xC, 2xC. The downward trend of the prediction error is less significant than with varying $N$.

## C   Additional Details on Design Choices

### C.1   Compute-FLOPs $C$ instead of $(N, D)$ for task loss prediction

We test if compute-FLOPs $C$ can be used directly for task loss prediction instead of $(N, D)$ in the over-trained regime with the ladder models. We fit a similar power function

$$L(C) = A/C^{\alpha} + E \tag{4}$$

where $A, \alpha, E$ are parameters to fit.

Figure 11 shows the function fitting and prediction. Compared with Figure 3, the relative fitting error is higher than using $(N, D)$ on all tasks, likely because Equation 4 has fewer free parameters than Equation 1. For the target models, the task loss prediction errors are generally worse than using $(N, D)$, including on MMLU; ARC-Challenge and ARC-Easy are the outliers (Table 3).

Overall, compute-FLOPs are not expressive enough to distinguish between compute-optimal and overtrained models with equal FLOPs but different losses (Hoffmann et al., 2022). Thus, we do not use it in our main method.

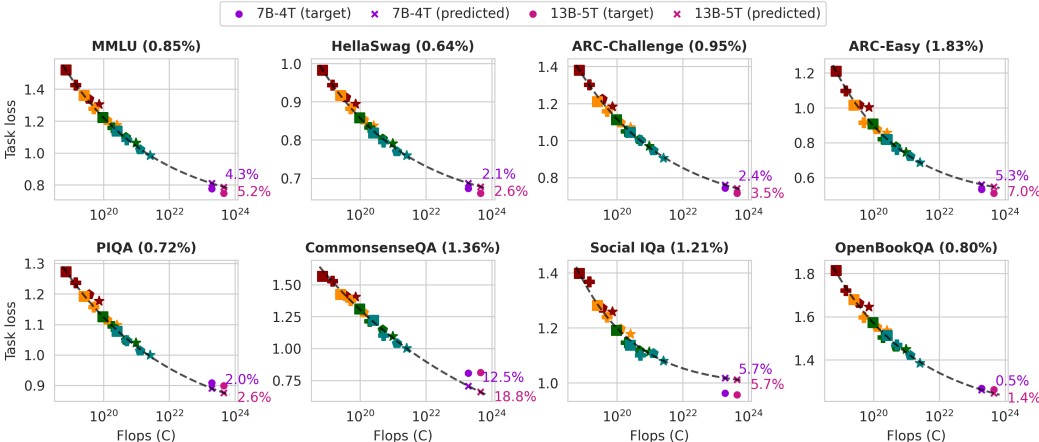

Figure 11: Step 1 using compute-flops $C$ instead of $(N, D)$. ■ = 1xC; ✚ = 2xC; ⬟ = 5xC; ★ = 10xC. We report the average relative fitting error in parentheses following the task name, and prediction error in the plot next to the target model point.

## C.2 Task cross-entropy as the intermediate feature

Task loss only considers the correct choice of each problem. However, task RC accuracy is determined by the losses of both correct and incorrect choices, which is a major challenge for predicting downstream performance (Schaeffer et al., 2024):

$$\text{Acc} = \frac{1}{N} \sum_{i=1}^{N} \mathbb{1} \left[ \arg\max_k \left( -L_k^{(i)} \right) = \hat{k}^{(i)} \right], \tag{5}$$

where $L_k^{(i)}$ is the loss over the $k$'th answer option of the $i$'th example and $\hat{k}^{(i)}$ is the label of the correct answer [5].

To account for incorrect answers, we define an alternative intermediate feature. We commonly maximize the accuracy by minimizing the task cross-entropy as a surrogate loss, in which the $L_k$ terms are used as logits:

$$\text{TaskCE} = \frac{1}{N} \sum_{i=1}^{N} \left( L_{\hat{k}^{(i)}}^{(i)} + \log \sum_k \exp \left( -L_k^{(i)} \right) \right). \tag{6}$$

We refer to this as TaskCE to distinguish it from the language modeling cross-entropy over tokens.

**Step 2 scaling with Task Cross-Entropy.** When inspecting the ladder models, we notice that the task cross-entropy correlates with accuracy in a more linear fashion than is the case for BPB. We therefore seek for a function $Acc(TaskCE)$ that is linear for large values of $TaskCE$, but for small values of $TaskCE$ should approach an asymptote of perfect accuracy (where all the probability mass is concentrated on the correct answer). A good candidate for this transition is the log-sigmoid function, leading us to conjecture the following formula:

$$Acc(TaskCE) = 1 - a \log \sigma \left( -k(TaskCE - TaskCE_0) \right), \tag{7}$$

where $\sigma(\cdot)$ is the sigmoid function, and $a, k, TaskCE_0$ are constants to be fit. This expression is not valid in the regime of large task cross-entropies, where models perform random guessing. Thus, we only use data points from the last 50% of training for curve fitting.

---

[5]Loss terms may include some normalization, e.g., by the number of characters in each answer or the unconditional answer probability (Gu et al., 2024).

|  | 7B-4T | | | | 13B-5T | | | |
|  | BPB | | TaskCE | | BPB | | TaskCE | |
|  | Error | %Error | Error | %Error | Error | %Error | Error | %Error |
|---|---|---|---|---|---|---|---|---|
| MMLU | 0.6 | 1.3% | 9.0 | 18.3% | 0.3 | 0.6% | 10.4 | 20.2% |
| HellaSwag | 1.2 | 1.4% | 5.9 | 7.2% | 2.1 | 2.5% | 8.7 | 10.5% |
| ARC-Challenge | 10.4 | 16.9% | 13.1 | 21.3% | 11.1 | 17.5% | 12.3 | 19.5% |
| ARC-Easy | 8.0 | 9.4% | 4.5 | 5.4% | 10.0 | 11.4% | 5.4 | 6.2% |
| PIQA | 0.8 | 1.0% | 2.5 | 3.1% | 0.9 | 1.1% | 2.4 | 2.9% |
| CommonsenseQA | 3.1 | 4.2% | 1.9 | 2.6% | 3.5 | 4.7% | 2.0 | 2.7% |
| Social IQa | 1.2 | 2.0% | 0.5 | 0.7% | 1.7 | 2.7% | 0.8 | 1.2% |
| OpenBookQA | 5.2 | 10.6% | 3.5 | 7.1% | 3.7 | 7.8% | 0.1 | 0.2% |

Table 6: Comparison of original and new prediction errors for 7B-4T and 13B-5T models across tasks. Absolute (Error) and relative (%Error) differences are shown.

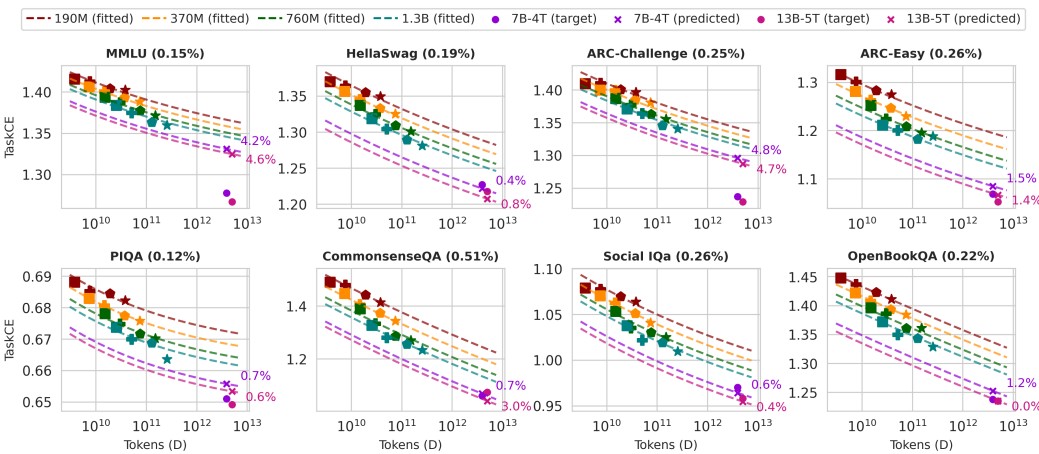

Figure 12: Predicting final *task cross-entropy* (Equation 6) from model parameters and token budget in step 1.

**Results.** Figure 12 and Figure 13 show the results of fitting step 1 and step 2 with the task cross-entropy as output and input variable, respectively. Figure 14 visualizes the predictions of combining these two steps, and Table 6 compares the predictions errors between using task cross-entropy or the correct answer loss as the intermediate feature. We make the following observations:

- In step 2, the task cross-entropy is overall a more accurate predictor of RC accuracy: The average prediction errors across 7B and 13B models is 2.75% vs. 3.6% for the correct answer loss in Figure 4. We note that the step 2 fit in Figure 13 is particularly good on ARC-Challenge and OpenBookQA, but substantially worse on HellaSwag. We note all our observations for HellaSwag are in the "linear" regime of the log-sigmoid, making it difficult to estimate the curvature of the transition to the horizontal asymptote.

- The average fitting and extrapolation errors in step 1 of Figure 12 are moderate and comparable to Figure 3. However, we notice that the fitting errors for TaskCE tend to be more skewed, i.e., the fitted Chinchilla curves tend to underestimate the performance of the 190M-1xC model and overstimate the 190M-10xC, while underestimating the 1B-1xC and overestimating the 1B-10xC. As these errors appear more systematic, it raises questions whether this parametric form is a good fit over much larger scales. Notably, it results in substantial errors in the task cross-entropy of MMLU and ARC-Challenge (4.2% - 4.8%).

- Finally, we note that the values for TaskCE fall into a narrow range. This means that the predictions in step 2 are highly sensitive to small changes in the task cross-entropy, e.g., a difference in task cross-entropy of less than 0.05 nats in PIQA can make the difference

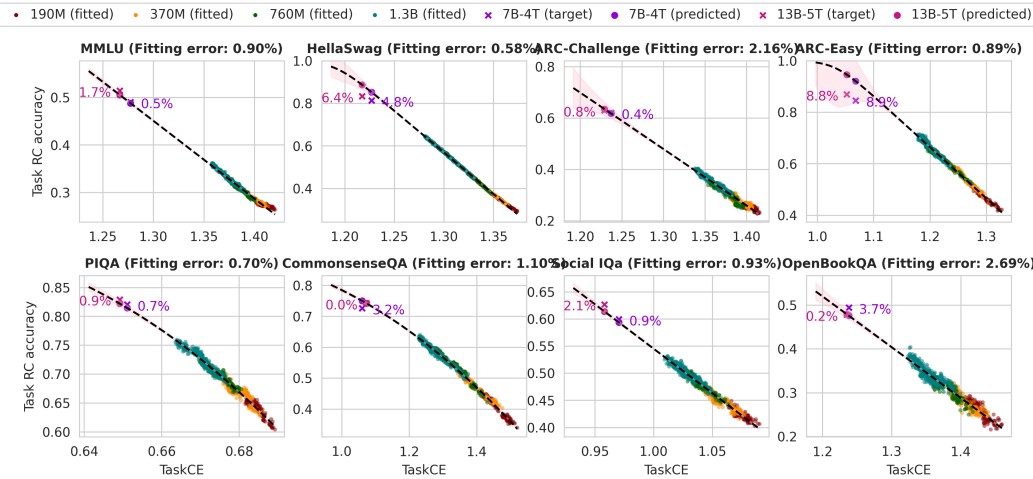

Figure 13: Predicting the task metric from the *task cross-entropy* (Equation 6) in step 2.

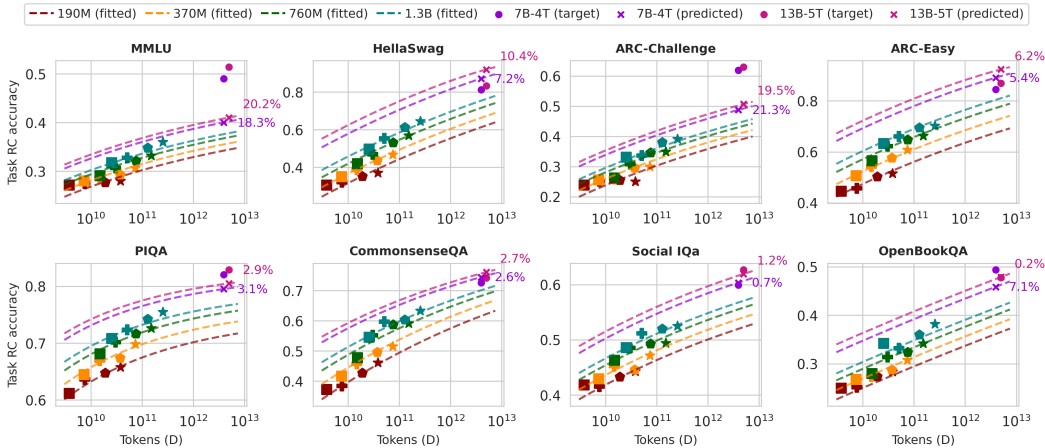

Figure 14: Chaining predictions from step 1 (Figure 12) and step 2 (Figure 13) with *task cross-entropy* as the intermediate feature.

between random task accuracy (at $TaskCE = 0.69$) and 83% accuracy (at $TaskCE = 0.65$). As a consequence, small relative errors in step 1 (0.6% for the 13B model on PIQA) are amplified in Step 2 and result in larger overall errors in accuracy (2.9% for PIQA), despite a good fit in step 2 (0.9% prediction error for the 13B accuracy). Similarly, a step-1 prediction error of 4.7% for the 13B model on ARC-Challenge results in an overall error of 19.5%.

## C.3 General language modeling loss as the intermediate feature

Language modeling loss (LM loss) on held-out sets has been shown to follow the power law (Hoffmann et al., 2022). Here, we consider if we can map it to task performance.

We experiment with using the LM loss on the C4-en validation set (Raffel et al., 2019) as the intermediate feature.

Figure 15 shows the function fitting and predictions.

In step 1, the fitted function underestimates the C4 loss by 2.0% for 7B-4T and 3.8% for 13B-5T. This error is higher than that on task loss prediction for 4 out of 8 tasks. The step 2

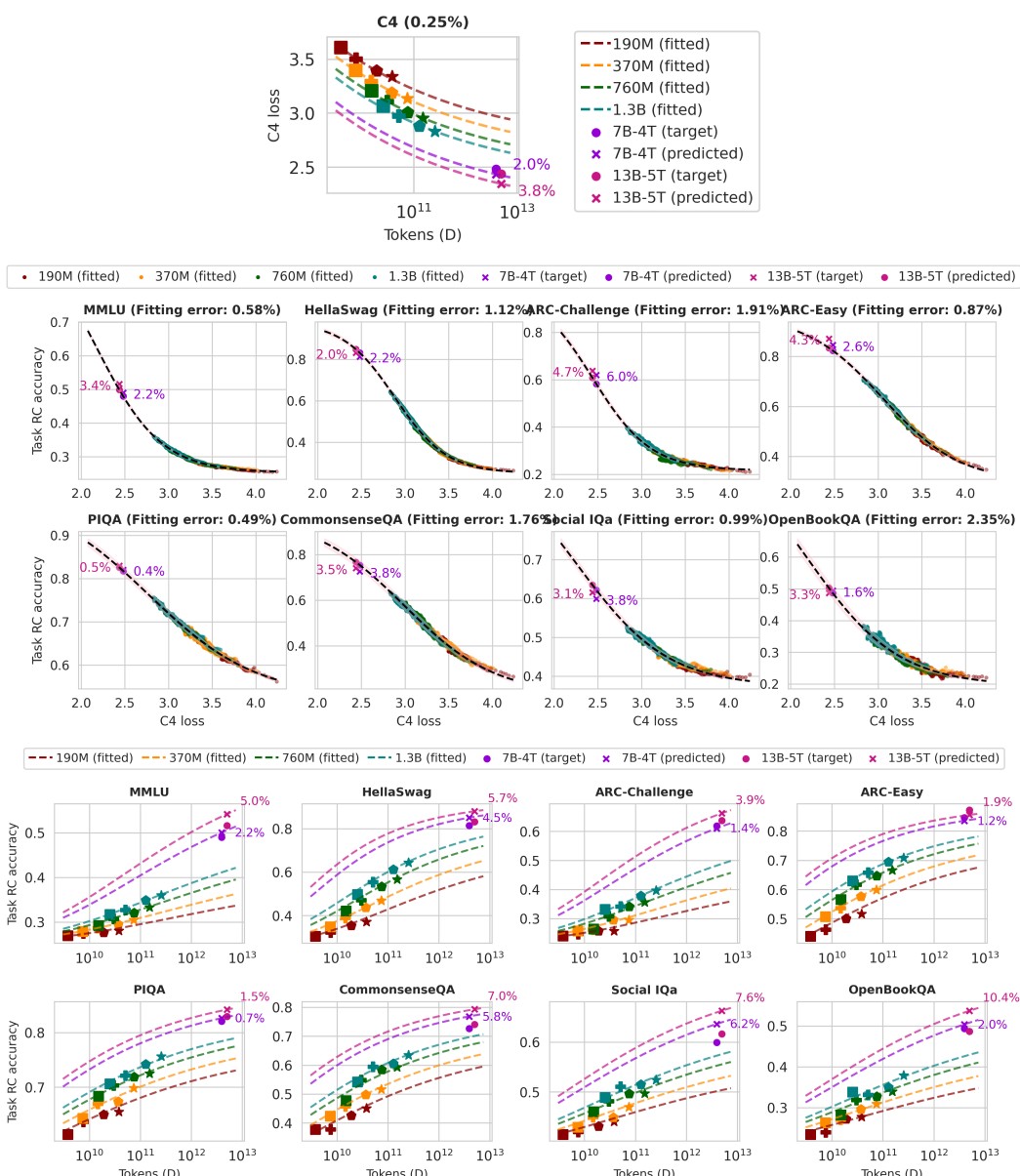

Figure 15: Using language modeling loss on C4-en validation as the intermediate feature. **From top to bottom**: step 1, step 2, and chaining the two steps.

error is also higher than using task loss on 4 out of 8 tasks. When chaining the two steps to predict task accuracy, using C4 loss as intermediate feature resulted in higher error on 5 tasks – MMLU, HellaSwag, CommonsenseQA, Social IQA, and OpenBookQA. Prediction error is lower on ARC-Challenge and ARC-Easy. We conclude that using C4 loss can benefit certain tasks where task loss does not work well.

## C.4 Comparison of three intermediate features

## C.5 Combining step 1 and 2 into a single step

Here, we try to directly predict task accuracy from the training scale $(N, D)$ in one step, by combining Equation 1 and Equation 2 into a single parameterized function, and merging

| 7B-4T Model - Absolute Errors | | | | | |
|---|---|---|---|---|---|
| **Task Name** | **Examples** | **Task Loss** | **C4 Loss** | **TaskCE Loss** | **Row Average** |
| MMLU | 14042 | 0.6 | 1.1 | 9.0 | 3.57 |
| HellaSwag | 10042 | 1.2 | 3.7 | 5.9 | 3.6 |
| ARC-Challenge | 1172 | 10.4 | 0.9 | 13.1 | 8.13 |
| ARC-Easy | 2376 | 8.0 | 1.0 | 4.5 | 4.5 |
| PIQA | 1838 | 0.8 | 0.6 | 2.5 | 1.3 |
| CommonsenseQA | 1221 | 3.1 | 4.2 | 1.9 | 3.07 |
| Social IQa | 1954 | 1.2 | 3.7 | 0.5 | 1.8 |
| OpenBookQA | 500 | 5.2 | 1.0 | 3.5 | 3.23 |
| **Average** | - | **3.61** | **2.17** | **5.34** | **3.71** |

| 13B-5T Model - Absolute Errors | | | | | |
|---|---|---|---|---|---|
| **Task Name** | **Examples** | **Task Loss** | **C4 Loss** | **TaskCE Loss** | **Row Average** |
| MMLU | 14042 | 0.3 | 2.6 | 10.4 | 4.43 |
| HellaSwag | 10042 | 2.1 | 4.7 | 8.7 | 5.17 |
| ARC-Challenge | 1172 | 11.1 | 2.5 | 12.3 | 8.63 |
| ARC-Easy | 2376 | 9.9 | 1.7 | 5.4 | 5.67 |
| PIQA | 1838 | 0.9 | 1.2 | 2.4 | 1.5 |
| CommonsenseQA | 1221 | 3.5 | 5.2 | 2.0 | 3.57 |
| SocialIQa | 1954 | 1.6 | 4.7 | 0.8 | 2.37 |
| OpenBookQA | 500 | 3.8 | 5.1 | 0.1 | 3.0 |
| **Average** | - | **4.2** | **3.23** | **6.0** | **4.48** |

| 7B-4T Model - Relative Errors (%) | | | | | |
|---|---|---|---|---|---|
| **Task Name** | **Examples** | **Task Loss** | **C4 Loss** | **TaskCE Loss** | **Row Average** |
| MMLU | 14042 | 1.3% | 2.2% | 18.3% | 7.27% |
| HellaSwag | 10042 | 1.4% | 4.5% | 7.2% | 4.37% |
| ARC-Challenge | 1172 | 16.9% | 1.4% | 21.3% | 13.2% |
| ARC-Easy | 2376 | 9.4% | 1.2% | 5.4% | 5.33% |
| PIQA | 1838 | 1.0% | 0.7% | 3.1% | 1.6% |
| CommonsenseQA | 1221 | 4.2% | 5.8% | 2.6% | 4.2% |
| Social IQa | 1954 | 2.0% | 6.2% | 0.7% | 2.97% |
| OpenBookQA | 500 | 10.6% | 2.0% | 7.1% | 6.57% |
| **Average** | - | **5.17%** | **3.14%** | **8.37%** | **5.56%** |

| 13B-5T Model - Relative Errors (%) | | | | | |
|---|---|---|---|---|---|
| **Task Name** | **Examples** | **Task Loss** | **C4 Loss** | **TaskCE Loss** | **Row Average** |
| MMLU | 14042 | 0.7% | 5.0% | 20.2% | 8.63% |
| HellaSwag | 10042 | 2.5% | 5.7% | 10.5% | 6.23% |
| ARC-Challenge | 1172 | 17.5% | 3.9% | 19.5% | 13.63% |
| ARC-Easy | 2376 | 11.4% | 1.9% | 6.2% | 6.5% |
| PIQA | 1838 | 1.1% | 1.5% | 2.9% | 1.83% |
| CommonsenseQA | 1221 | 4.7% | 7.0% | 2.7% | 4.8% |
| Social IQa | 1954 | 2.7% | 7.6% | 1.2% | 3.83% |
| OpenBookQA | 500 | 7.8% | 10.4% | 0.2% | 6.13% |
| **Average** | - | **5.8%** | **4.66%** | **9.03%** | **6.49%** |

Figure 16: Comparison of absolute and relative prediction errors for all three intermediate features. Using C4 as the intermediate works well for certain tasks, but results in overall higher errors. Task loss and TaskCE loss perform comparably. However, MMLU and HellaSwag have higher number of instances, and task loss performs better for them.

parameters $k$ and $L_0$ into $A, B, E$, so that we reduce to 7 free parameters:

$$Acc(N, D) = \frac{a}{1 + \exp\left(-(A/N^\alpha + B/D^\beta + E)\right)} + b \tag{8}$$

With this, we remove the *a priori* definition of a specific intermediate feature (i.e., the $A/N^\alpha + D/D^\beta + E$ expression no longer carries a specific meaning), while preserving the representation power of the function. This function can be harder to fit, since it has more free parameters, is not convex, and we cannot use any data points collected from intermediate checkpoints as we did in step 2.

We fit this function with data points from the final checkpoints of the ladder models, using the same optimization method as in step 1 (Figure 17). The prediction error is higher on 4 out of 8 tasks – MMLU, ARC-Challenge, CommonsenseQA, and Social IQA. In particular, on CommonsenseQA the fitted functions does not vary with respect to model size, indicating a degenerated function fitting. We conclude that the single-step approach is not as robust as the two-step approach.

# D  Scaling law predictions for 32B-6T

We demonstrated our method on 7B-4T and 13B-5T models, using only 1% of the compute required for the target models. Here, we test the robustness of the ladder, by trying to

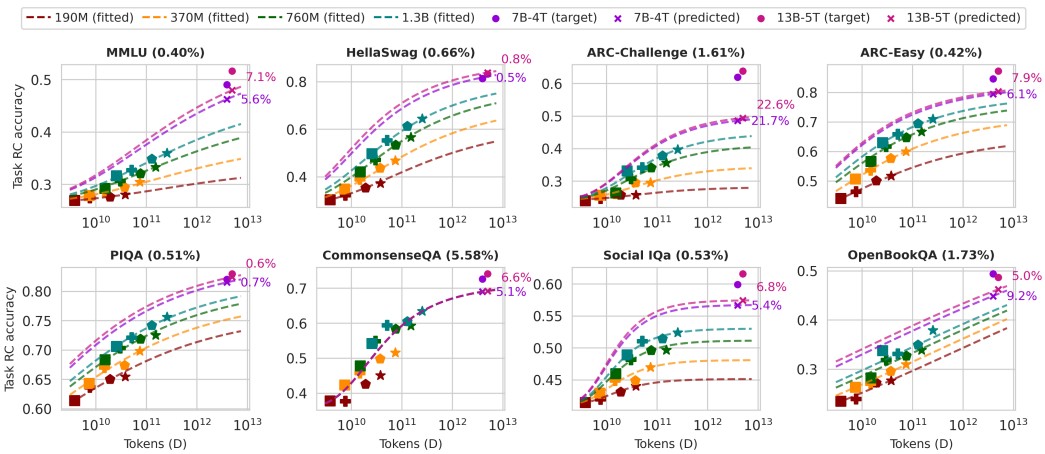

Figure 17: Task RC accuracy vs training scale $(N, D)$, with fitting on the single-step function in Equation 8.

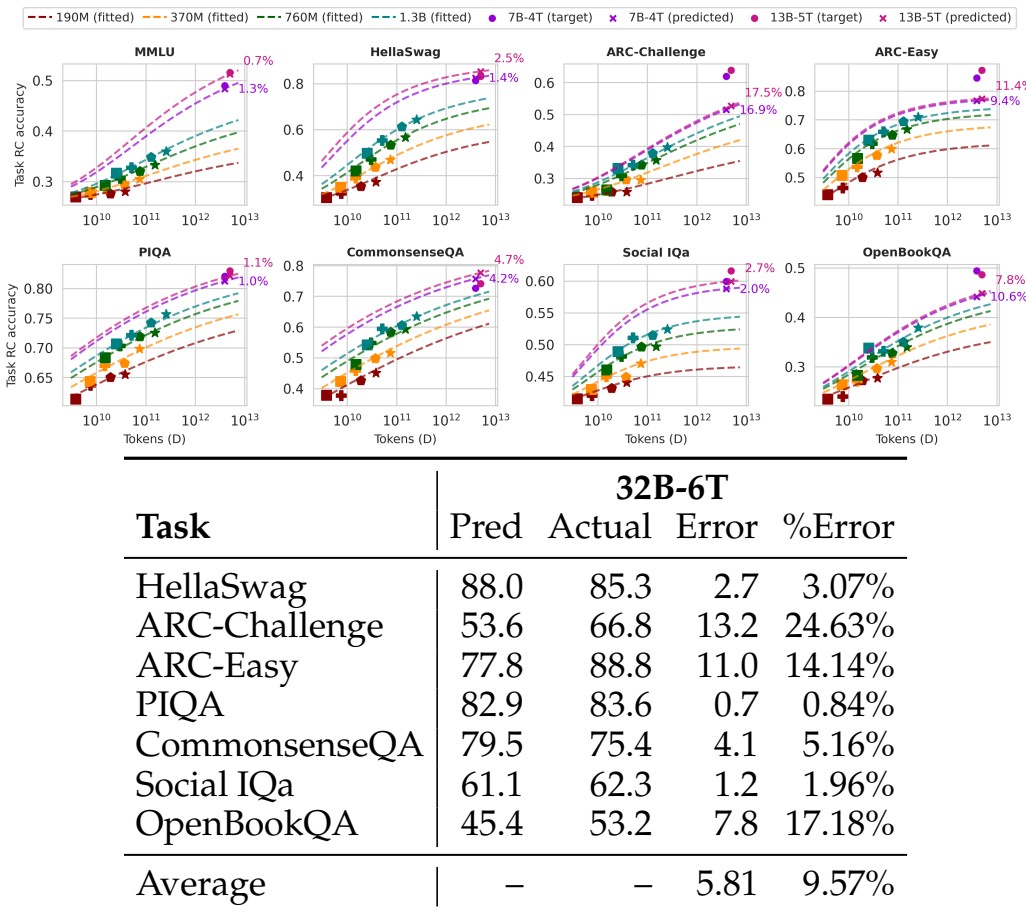

|  | **32B-6T** | | | |
| **Task** | Pred | Actual | Error | %Error |
| HellaSwag | 88.0 | 85.3 | 2.7 | 3.07% |
| ARC-Challenge | 53.6 | 66.8 | 13.2 | 24.63% |
| ARC-Easy | 77.8 | 88.8 | 11.0 | 14.14% |
| PIQA | 82.9 | 83.6 | 0.7 | 0.84% |
| CommonsenseQA | 79.5 | 75.4 | 4.1 | 5.16% |
| Social IQa | 61.1 | 62.3 | 1.2 | 1.96% |
| OpenBookQA | 45.4 | 53.2 | 7.8 | 17.18% |
| Average | – | – | 5.81 | 9.57% |

Figure 18: Task accuracy prediction for the 32B model using task loss as the intermediate feature. ■ = 1xC; ✚ = 2xC; ⬟ = 5xC; ★ = 10xC. Prediction error is next to the target.

predict the accuracies for a much larger model of the same family: OLMo 2 32B, trained to 6T tokens (32B-6T). This model was trained with $1.15 \times 10^{24}$ FLOPs, and the ladder models account for only 0.45% of the compute of this model.

Figure 18 shows the predictions and errors for 32B-6T model. As expected, and following from B.3, the overall prediction error is higher We also observe a similar trend of higher errors for high variance tasks like ARC-C and ARC-E as with the smaller target models. However, for lower-variance tasks such as HellaSwag, PiQA, and SocialIQA, we can still predict the accuracy within an absolute error of 3 points. This indicates that the ladder models are fairly robust in their prediction abilities even as the target model is scaled to be much larger.

