# OpenReview forum: "Establishing Task Scaling Laws via Compute-Efficient Model Ladders"
_colmweb.org/COLM/2025/Conference — COLM 2025_

### Official Review · Reviewer_s6ea · 2025-05-06

**Rating:** 7
**Confidence:** 4
**Ethics Flag:** 1

**Summary:**

The paper introduces *model ladders*, a two-step framework that fits power-law surfaces and then maps them to downstream task accuracy for larger language models. By training a fixed suite of small “ladder” models, the authors can forecast the accuracy of much larger targets (7 B and 13 B parameters) while spending only a fraction of the compute. Their approach predicts downstream scores to within ~4 % relative error and is validated across eight diverse language-understanding benchmarks. Overall, the work offers a fresh perspective on scaling laws by directly linking small-model experiments to practical accuracy estimates for significantly larger systems.

**Questions To Authors:**

1. Have you attempted to predict larger-parameter checkpoints? If not, what evidence suggests the fitted exponents (α, β) remain stable?
2. Could the same ladder (trained on OLMo architecture) predict performance of a different LM family model if both share token budgets?

**Reasons To Accept:**

The paper seems to offer a novel perspective towards scaling laws in general, where instead of just predicting the LM loss, there was an attempt to fit the accuracy function. The study spans 8 diverse tasks, ablation over three intermediates, two moderately large LMs, FLOPs-only baselines, a single-step variant, and detailed checkpoint variance analysis. It is without doubt an empirically rigorous work.

**Reasons To Reject:**

There were some assumptions made and some result reported in the paper which might require a bit more justification -
1. For four tasks, the test sets were used, and for the other four, the validation set. I am assuming one or the other wasn't available for all which prompted the authors to make this choice, but it should be explained in the paper.
2. The training converged after 4T in 7B and went up to 5T in 13B. Why?
3. The two-step model plus per-task fits introduces many degrees of freedom; although L-BFGS is used, the risk of local minima or over-parameterization is not analyzed.

---

> ### Author Response · Authors · 2025-06-03
>
> Thanks for acknowledging our rigorous experiments. Below we address your comments:
> 1. **Test vs validation set:** As we mentioned in Sec 2.3, we use test sets whenever possible, and fallback to validation set when the test set (or the labels) is not available.
> 2. **Convergence:** The training data sizes do not indicate convergence. For the target models, we simply used the final pre-training checkpoints of OLMo, which is 4T tokens for the 7B model and 5T tokens for the 13B model. These are choices made by the OLMo model developers.
> 3. **Over-parameterization:** We ablated a functional form with fewer free parameters (7 instead of 9) in Appendix C.5. However, this led to degeneration of function fitting on some tasks, so we concluded that a bit of over-parameterization can help make our method more robust.
> 4. **Predicting for larger models:** A 32B version of OLMo 2 was released after we submitted the paper, and below is our prediction of its task performance (using the same set of fitted scaling law parameters as in our submission):
> | Task | Pred | Actual |
> | --- | --- | --- |
> | HellaSwag | 88.0 | 85.3 |
> | ARC-Challenge | 53.6 | 66.8 |
> | ARC-Easy | 77.8 | 88.8 |
> | PIQA | 82.9 | 83.6 |
> | CSQA | 79.5 | 75.4 |
> | Social IQA | 61.1 | 62.3 |
> | OpenBookQA | 45.4 | 53.2 |
>
> 	Our predictions are accurate on tasks like HellaSwag, PIQA, and Social IQA, which are all within an absolute error of 3 points (similar to the predictions for 7B and 13B models). The predictions are off on ARC due to the chaining error in two steps.
> 5. **Transferring to another LM family:** We don't think the OLMo ladders can be directly applied to predict for another LM family, because differences in training data mix, model architecture, etc can shift the parameters of scaling laws.

---

### Official Review · Reviewer_Dg36 · 2025-05-07

**Rating:** 7
**Confidence:** 4
**Ethics Flag:** 1

**Summary:**

This paper presents an effective method to estimate the benchmark performance of LLMs, moving beyond previous work that only focuses on the pre-training loss as a target metric. The authors conduct comprehensive experiments to demonstrate the effectiveness of the proposed approach and use extended analysis to verify several design choices.

**Questions To Authors:**

N/A

**Reasons To Accept:**

1. This paper focuses on a very important research topic. The paper writing is easy to follow, and the presentation is wonderful.

2. For the method part, the proposed algorithm is sound and effectively addresses the challenges in estimating the benchmark performance.

3. For the experiments part, the setting is reasonable, and the results are impressive. The further analysis is extensive and interesting.

**Reasons To Reject:**

Major concern:

This paper is overall a very good paper, regarding the research problem, the method effectiveness, experiments, writing, and the figures/tables/results presentation. The only major concern is that the proposed method is almost the same as the one mentioned in the Llama-3 technical report [1]. And according to the arxiv date, there is a roughly 5-month gap, so they should not be considered as concurrent work. The authors address this by saying, "Unlike our work, which uses a fixed ladder of small 261 models, they rely on first finding compute-optimal models." and "Prior work has shown accuracy for only one specific task (ARC-Challenge).". These two statements, at least for my understanding, cannot distinguish the unique difference, since (1) Llama-3-series is also over-trained, so the compute-optimality may not be an important design choice; (2) You cannot imagine Meta only tries on a single benchmark for the scaling laws, alghouthg they are only presenting one in the paper.

Generally, this paper lacks a unique contribution and novelty as a good research paper. But this paper provides new, comprehensive details, including the concrete implementation, design choices, and large-scale verification of an existing, well-established "closed-source" method. So it's a very good technical report. I think the insights/details provided in this paper are valuable and useful, so I vote for acceptance. But the AC/SAC may consider the raised concern if COLM prefers papers with unique contributions and novel ideas.


Minor:
It would be interesting to see the comparison with the two-stage approach proposed in [2] to decide which is the best intermediate metric to keep track of (pre-training loss vs. task loss). But I totally understand if such a comparison cannot be achieved due to resource limitations. So this is a minor point.


[1] The Llama 3 Herd of Models. Aaron Grattafiori et al.

[2] Scaling laws for predicting downstream performance in llms. Yangyi Chen et al.

---

> ### Author Response · Authors · 2025-06-03
>
> Thanks for acknowledging our contribution to opening up an important method in understanding language models. Also thanks for recognizing that our approach is sound and effective for addressing the problem. Below we address your concerns:
> 1. Comparison with the Llama 3 paper:
>     * **Comprehensiveness of benchmarks, and analyses:** While we believe Llama 3 tried more benchmarks, we don't know how well their method works there because they didn't report them. In contrast, we report on a suite of 8 tasks in our paper, including tasks where our predictions weren't very good and we investigated the reasons (Sec 5). We hope this can provide the broader community more objective insights to this approach.
>     * **Fully open and reproducible:** As acknowledged by the reviewer, this paper contributes to the science of open language model training. We documented our method and setup to a great level of details. In contrast, the Llama 3 paper leaves out many important details – such as the formal definition of the intermediate feature, the functional form and optimization algorithm for their "step 2" – making reproducibility challenging. uncovering how scaling laws can be established for language model training, and what properties tasks should have to be used for scaling laws, and provides detailed analyses since data and details of model training are clear. In addition, many details were missing from the Llama 4 paper, which reproducibility challening.
>     * **Over-trained vs. compute-optimal:** The Llama 3 paper only reports their scaling law results on a 405B model for 15T tokens, which is in the compute-optimal regime according to the Chinchilla paper. It does not report results for their over-trained models, including their 8B nor 70B.
> 2. **Pre-training loss as intermediate feature:** We ablated this, with a similar pre-training loss (the LM loss on the C4 validation set). Our discussion can be found in Sec 2.2, Sec 4, and Appendix C.3.

---

> > ### Comment · Reviewer_Dg36 · 2025-06-05
> >
> > Thanks for the additional clarification. Slightly raise my score.

---

### Official Review · Reviewer_ExxQ · 2025-05-13

**Rating:** 6
**Confidence:** 4
**Ethics Flag:** 1

**Summary:**

The authors develop a means of estimating task-specific LLM performance as a function of model and dataset size. In theory, this enables estimating the performance of large models trained on large dataset on specific downstream tasks, without actually training them, by extrapolating the scaling laws. The evaluation is focused on predicting performance on tasks in the ranked classification format, which simplifies evaluation because candidate answers can be scored directly, instead of requiring models to select a specific answer from a list of candidates, which smaller models cannot do effectively. To predict the task-specific loss, two functions are fit -- one to approximate the task specific loss as a function of model and dataset size, and the second to predict the accuracy as a function of the predicted loss,  thus the predicted task-loss is treated as an "intermediate feature". The evaluation is very thorough and the presented results are compelling, but because there is no benchmark or baseline for this setting, it's difficult to know how compelling the results actually are.

**Reasons To Accept:**

- clearly written paper with very thorough exposition and evaluation of task-specific accuracy prediction using ladder models to estimate performance of larger models.

- clear applications to foundation and larger model training, especially in the context of planning.

- simple implementation that should be easy to replicate or extend

**Reasons To Reject:**

- lack of baselines makes empirical results difficult to interpret -- no discussion of alternative approaches.

- limited to ranking-style tasks in its current formulation, does not work for many tasks because the approach relies upon the log-loss of the model, which may not be enough to connect to task accuracy for many settings.

- limited discussion of real-world applications and limitations

---

> ### Author Response · Authors · 2025-06-03
>
> Thanks for acknowledging our thorough evaluation and practical relevance to large model training. Below we address your concerns:
> 1. **Lack of enough baselines:** We have ablated many alternative design choices, including those proposed in prior work. These include alternative intermediate features (as in Gadre et al., 2024) and alternative input features (as in Dubey et al., 2024). Comparison to these baselines can be found in Sec 4.
> 2. **Limited to ranking-style tasks:** We agree, and we acknowledged extending task format as potential future work in Sec 8.
> 3. **Real-world applications:** A real-world application of our method is for LM developers to make decisions before kicking off big pre-training runs. With the predicted task performance, developers can decide if it aligns with their expectations and if not, iterate on the data and training settings w/o committing massive amounts of compute. This can make the model developing workflow more efficient and save cost.

---

> > ### Comment · Reviewer_ExxQ · 2025-06-09
> >
> > thanks for your response, i'm maintaining my score of 6

---

### Official Review · Reviewer_Jq1a · 2025-05-14

**Rating:** 5
**Confidence:** 3
**Ethics Flag:** 1

**Summary:**

The paper proposes the use of smaller capacity ladder models (ranging between 190M to 1.3B parameters) to predict the downstream task performance of larger (7B and 13B parameter) OLMo target models over a selection of tasks from the OLMES evaluation suite.
The investigation takes into consideration the number of model parameters (N) and the amount of training data (D) used for the creation of the models.

The proposed approach involves two stages.
The first one tries to predict the task-specific loss based on the parameters N and D, producing so-called intermediate features.
During the second stage, the task loss is used for predicting the downstream task performance.

The approach is easy to follow, but the raises several question mostly due to insufficient comparison to alternative approaches that are elaborated among the weaknesses in more detail.

**Questions To Authors:**

The paper claims that recent LMs are overtrained (l. 54), but it also mentions that one of the target OLMo models was __only__ trained over 3.9T tokens (l. 77).
The latter statement feels contradictory to the former one.
Also, the fact that the small amount of training data was meant to be mitigated by adding 50B  (0.05T) extra training tokens, increasing the 3.9T tokens to 3.95T feels insufficient.

**Reasons To Accept:**

Predicting task performance of LLMs is a timely chosen topic and the writing is generally easy to follow.

**Reasons To Reject:**

The paper lists most of the relevant related work, but does not include a meaningful comparison to the related approaches.
For instance, the related work section claims that ``Chen et al. (2024) also employs a two-stage approach for predicting downstream performance, but uses the pre-training loss instead of a task-specific loss as the intermediate step.'' (line 261--263), but there is no empirical evidence presented that the latter approach is more performant and there is no discussion on the pros and cons of the different approaches.

Even though the amount of compute required to train the ladder models is marginal compared to the costs of training the target model(s), it is not negligible in absolute terms.
The amount of compute used for pre-training the additional models, and extracting the intermediate features from them could just be used in a variety of alternative ways for giving a task performance estimate (e.g. evaluating the target models on a sample of the downstream task from time to time).
The paper does not convincingly demonstrate why the proposed approach is superior to such alternatives.

As LLMs are known to be highly sensitive to prompt wording it is debatable to what extent task performance prediction is feasible as using slightly different prompts could result in radically different task performance.

Experiments are only performed on a single model family.

---

> ### Author Response · Authors · 2025-06-03
>
> Thanks for acknowledging that the paper is well written and timely for a critical problem in language model training. Below we address your comments:
> 1. **Comparison with related work, such as pre-training loss as intermediate feature:** We indeed ablated this in our experiments (please check our discussion is in Sec 2.2, Sec 4, and Appendix C.3) with a similar pre-training loss (the LM loss on the C4 validation set). We found that task loss leads to more accurate predictions for benchmarks with low variance, and LM loss works better for high-variance benchmarks.
> 2. **Monitoring performance of intermediate checkpoints instead of establishing scaling laws:** We highlight that establishing scaling law using our method can offer a prediction before kicking off the big pre-training runs, so LM developers can iterate on their recipe w/o committing massive amounts of compute. For example, training a 7B and 13B model to 200B tokens (a point where we can start making rough extrapolations) would cost $2.4 \times10^{22}$ flops, or **over 4 times** of that for training the ladder models. If at this point we decide that the model will not be good enough, we would need to start over and all such compute is wasted. Monitoring performance of intermediate checkpoints is still a good sanity check during big training runs, but there exists no principled way to extrapolate task learning curves, especially given the complication of learning rate schedule.
> 3. **Sensitivity to prompts:** To prevent the impact of prompts, we followed the standards introduced in the OLMES paper, which aimed at standardizing prompts for language model evaluation such that they are less susceptible to changes in prompts. We leave varying prompts and model families as future work.
> 4. **Claim about current LMs are over-trained:** We refer to over-training with respect to the compute-optimal number of tokens as suggested by the Chinchilla paper. Our target models are already heavily over-trained. For example, the 7B model trained on 3.95T tokens is 28x the compute-optimal as suggested by the Chinchilla paper. The extra 50B tokens at the end were **not** meant for adding more training data, but for completing the LR decay schedule (which makes a big difference when not complete).

---

> > ### Comment · Reviewer_Jq1a · 2025-06-11
> > **Thank you**
> >
> > Thanks for the clarifications.
> > Considering that the findings were only validated for a single model family, it remains a question how well the recommendations formulated in the paper generalize to further and more popular LMs.
> > To this end, I maintain my original evaluation.

---

> ### Comment · Area_Chair_se41 · 2025-06-05
>
> Hi Reviewer Jq1a! Just a reminder that the discussion period for COLM papers has begun. Could you please take a look at the author response to review and let them know whether it addresses any of your outstanding questions?
>
> Thanks,
> Your AC

---

### Decision · Program_Chairs · 2025-07-08

**Decision:**

Accept

**Comment:**

This paper describes a method for estimating LM performance on downstream tasks by first (1) estimating a standard scaling law for a task-specific loss function, then (2) learning a translation from task losses into classification accuracies. Reviewers agree that this is an important topic and that the method is simple and convincingly evaluated. There were some concerns about related work: (1) methods that use alternative intermediate features, which the authors point out they already compare to (though Tab 3 could make this clearer); and (2) similar ideas in the most recent Llama system description paper, which I think make an open study like the present paper more valuable.